# Development of RNAi Methods for the Mormon Cricket, *Anabrus simplex* (Orthoptera: Tettigoniidae)

**DOI:** 10.3390/insects13080739

**Published:** 2022-08-17

**Authors:** Toan Hoang, Bert Foquet, Seema Rana, Drew W. Little, Derek A. Woller, Gregory A. Sword, Hojun Song

**Affiliations:** 1Department of Entomology, Texas A&M University, College Station, TX 77843, USA; 2Department of Biological Sciences, Illinois State University, Normal, IL 61790, USA; 3Department of Biological Sciences, University of Wisconsin-Milwaukee, Milwaukee, WI 53211, USA; 4USDA-APHIS-PPQ-Science & Technology-Insect Management and Molecular Diagnostics Laboratory (Phoenix Station), Phoenix, AZ 85040, USA

**Keywords:** Mormon cricket, RNAi, dsRNA, Orthoptera

## Abstract

**Simple Summary:**

Mormon crickets (*Anabrus simplex*) are native to the United States and can become devastating rangeland pests. These insects can form migratory bands numbering in the millions, which can traverse the western U.S., damaging agricultural crops, invading urban areas, and producing hazardous slicks on roadways. Current methods of managing these insects involve targeted applications of non-specific chemical insecticides, which can potentially have negative effects on the environment. In this study, we took the first steps toward developing RNA interference (RNAi) methods, a type of gene silencing technique that has potential for pest management, for Mormon crickets as a potential alternative to traditional broad-spectrum insecticides. We first generated genomic resources to identify several candidate genes and developed double-stranded RNA (dsRNA) for these genes, which could be injected into the insect to induce gene silencing. We performed RNAi experiments to characterize gene knockdown efficiencies and demonstrated that it is possible to elicit RNAi responses in the Mormon cricket by injection, but knockdown efficiencies varied according to the target genes and tissue types. We conclude that RNAi silencing is possible in Mormon cricket, but more work needs to be done before it can be effectively used as a population management method.

**Abstract:**

Mormon crickets are a major rangeland pest in the western United States and are currently managed by targeted applications of non-specific chemical insecticides, which can potentially have negative effects on the environment. In this study, we took the first steps toward developing RNAi methods for Mormon crickets as a potential alternative to traditional broad-spectrum insecticides. To design an effective RNAi-based insecticide, we first generated a de novo transcriptome for the Mormon cricket and developed dsRNAs that could silence the expression of seven housekeeping genes. We then characterized the RNAi efficiencies and time-course of knockdown using these dsRNAs, and assessed their ability to induce mortality. We have demonstrated that it is possible to elicit RNAi responses in the Mormon cricket by injection, but knockdown efficiencies and the time course of RNAi response varied according to target genes and tissue types. We also show that one of the reasons for the poor knockdown efficiencies could be the presence of dsRNA-degrading enzymes in the hemolymph. RNAi silencing is possible in Mormon cricket, but more work needs to be done before it can be effectively used as a population management method.

## 1. Introduction

Mormon crickets, *Anabrus simplex* Haldeman, 1852 (Orthoptera: Tettigoniidae), are native to North America and can, on occasion, become devastating and irritating rangeland pests in several western U.S. states (Figure 1) [1,2,3]. Under outbreak conditions, these insects can form enormous migratory bands containing billions of voracious individuals that move cohesively across the landscape [3,4,5,6,7]. Specific mechanisms of how outbreaks can form are currently unknown, but previous studies have shown that the outbreaking populations in the western U.S. are genetically distinct from the eastern non-outbreaking populations [8,9]. They are capable of causing major economic damage in terms of crop loss and rangeland destruction, typically in Colorado, Idaho, Montana, Nevada, Oregon, Utah, and Wyoming [10,11], though their status as pests can be exacerbated by other factors, such as drought and overgrazing [12]. If left unchecked, native rangelands damaged by outbreak populations can be overwhelmed by invasive plant species that replace the productive range with less desirable plants [13]. Outbreaks can even cause dangerous driving conditions on major roadways when the bands are crushed by passing vehicles, creating highly slippery conditions [4]. Farmers and ranchers have battled Mormon crickets since the pioneer days, and various cultural and chemical management measures have been employed [2] but often ineffectively.

Federal programs (e.g., the United States Department of Agriculture (USDA) Animal Plant Health Inspection Service (APHIS) Plant Protection and Quarantine (PPQ) Rangeland Grasshopper and Mormon Cricket Suppression Program), along with State, Tribal, and private land managers, have gone to great lengths to ensure that Mormon cricket populations remain below an economic impact threshold [10]. They have developed techniques to suppress populations before they reach outbreak levels, mainly via the use of baits (e.g., carbaryl, a nervous system interrupter) [14] and liquid insecticides (e.g., diflubenzuron, an insect growth regulator) [10]. However, these traditional methods are not always effective or desired for use, especially in areas of sensitive habitat and near species of concern [15,16,17]. In recent years, some efforts have been made to try biological control methods to manage these insects, as the desire for non-chemical alternative population management measures has become increasingly important and desired [11,18]. However, so far, neither protozoan parasites (*Nosema locustae*) [19] nor entomopathogenic fungi (*Beauveria bassiana* and *Metarhizium brunneum*) have been shown to be effective [11,20].

Ongoing advances in genomics and biotechnology provide new opportunities to develop innovative and potentially species-specific tools to manage insect pests [21,22,23,24]. One such method is the use of RNA interference (RNAi), which is a method of post-transcriptional gene silencing that originates in the innate immune response of organisms to RNA viruses [25]. RNAi is a conserved and naturally occurring defense mechanism in eukaryotes in which double-stranded RNAs (dsRNAs) downregulate the expression of a target gene in a sequence-specific manner [25,26]. The Dicer-2 enzyme, part of the RNase III endonuclease family, cleaves long dsRNAs into small interfering RNAs (siRNAs) [27], typically 19–21 base pairs (bp) long [28], and then forms a complex with the RNA Induced Silencing Complex (RISC) with the help of dsRNA binding protein, R2D2 [29]. Within the complex, each siRNA strand is unwound to release the passenger strand, and the remaining guide strand, with the use of Argonaute protein (Ago2) [30], cleaves the complementary mRNA, thus downregulating the expression of the target gene [31]. Originally described in the nematode (*Caenorhabditis elegans*) [26], RNAi has been found in all eukaryotes that have been investigated and has been established as a powerful tool for molecular biologists to study gene function and regulation [25,32,33]. The ability of RNAi to knockdown the expression levels of any target gene has quickly led scientists to apply this technique to insect pest management [23,34,35,36]. The premise is simple and powerful. Target genes in a particular insect pest that are critical for metabolism, development, or reproduction can be identified, which can be knocked down through RNAi to cause reduced fitness, abnormal development, or death. The potential benefits of RNAi-based insecticides compared to conventional chemical insecticides are manifold. For example, RNAi is target-specific by design, which eliminates the impact on non-target species [21,35,36,37,38]. Furthermore, because it leads to only a temporary knockdown of gene expression during key developmental stages, it does not have long-term consequences after release, which could be problematic for more persistent solutions, such as gene drive systems [39]. A well-designed RNAi can have a dramatic response in a short period of time to a targeted pest with few non-target effects, which is a desirable trait for an effective insecticide [36,38]. However, there are still some obstacles to overcome in making them commercially viable and effective alternatives to traditional chemical insecticides [35,40].

RNAi is known to work particularly well for the insect order Orthoptera (grasshoppers, katydids, crickets, and their relatives), to which Mormon crickets belong, in which a systemic knockdown can be triggered by a simple injection of dsRNA into the hemolymph [41], a method that has been frequently used for studying functional genetics of locust phase polyphenism in the desert locust (*Schistocerca gregaria*) [41,42,43,44,45], the migratory locust (*Locusta migratoria*) [46,47,48,49,50,51], and the two-spotted cricket (*Gryllus bimaculatus*) [52,53,54]. Nevertheless, there has never been an effort to develop RNAi-based molecular insecticides for Mormon crickets or, for that matter, any of the major grasshopper pests in the rangelands of the western U.S. This is most likely because there are no genomic resources available for these insects and because there are very few research labs (university, government, industry) with the expertise and capacity to develop such methods for Orthoptera. Currently, beyond traditional chemical insecticide methods, there are no effective alternative management measures available to suppress outbreak populations. With the active development of RNAi-based insect management methods by many labs worldwide and the pressing need for alternative management measures, there is a particularly strong motivation to develop a novel molecular insecticide for Mormon crickets, which are currently outbreaking in several western states and have been annually for several years now.

In this study, we took the first steps toward developing RNAi methods for Mormon crickets as a potential alternative to traditional chemical insecticides. To alleviate the lack of genomic resources, we first generated a de novo transcriptome of the Mormon cricket, which was used to identify seven highly expressed genes that were used as target genes for developing RNAi. We then characterized the RNAi efficiencies by injection of gene-specific dsRNA and conducted real-time quantitative PCR (qPCR) experiments to quantify mRNA-level knockdown. We also characterized the time course of the RNAi response for a subset of these target genes and demonstrated the presence of dsRNA-degrading enzymes in the hemolymph that might affect RNAi efficiency.

## 2. Materials and Methods

### 2.1. Transcriptome Sequencing

Three female adult specimens of Mormon crickets were used for transcriptome sequencing, collected from an outbreak population in Idaho (Elmore County, ~17 miles NE of Mountain Home, 43°15′46.2″ N 115°31′00.9″ W (WGS84)) on 7 July 2018 by the USDA-APHIS-PPQ-S&T-IMMDL (Phoenix Station) Rangeland Grasshopper and Mormon Cricket Management Team (USDA Rangeland Unit). Total RNA was extracted from the entire head, the thorax without the legs, and the abdomen of each specimen. Tissues were homogenized in Trizol (ThermoFisher Scientific, Waltham, MA, USA) using MagNA Lyser Green Beads (Roche, Basel, Switzerland), and RNA was extracted using Trizol–chloroform extraction, followed by a clean-up with an RNeasy mini kit (Qiagen, Hilden, Germany). RNA concentrations were measured with a Denovix DS-11 spectrophotometer. A total of 9 separate RNA extractions were performed and sent to Novogene for library preparation, sequencing, transcriptome assembly, and annotation. RNA degradation and contamination were monitored on 1% agarose gels. RNA purity was checked using the NanoPhotometer^®^ spectrophotometer (IMPLEN, Westlake Village, CA, USA). RNA integrity and quantitation were assessed using the RNA Nano 6000 Assay Kit of the Agilent Bioanalyzer 2100 system (Agilent Technologies, Santa Clara, CA, USA). 1 µg RNA per sample was used as input material for the RNA sample preparations. Sequencing libraries were generated using NEBNext^®^ Ultra™ RNA Library Prep Kit for Illumina^®^ (NEB, Ipswich, MA, USA) following the manufacturer’s recommendations, and index codes were added to attribute sequences to each sample. In order to select cDNA fragments of preferentially 250~300 bp in length, the library fragments were purified with the AMPure XP system (Beckman Coulter, Brea, CA, USA), and library quality was assessed on the Agilent Bioanalyzer 2100 system. Subsequently, samples were sequenced on an Illumina HiSeq4000 platform, and paired-end reads were generated. Raw reads were filtered to remove reads containing adapters or reads of low quality (uncertain nucleotides constituting more than 10% of either read, or nucleotides with a Phred score <20 constituting more than 50% of the read). After quality control and data quality filtering, these raw reads from different tissues were pooled together to assemble a de novo transcriptome using Trinity [55], with min_kmer_cov set to 2 and all other parameters set as the default. The reads were then filtered and clustered by Corset (v1.05) [56] to remove redundancies, and the longest transcripts per cluster were selected as unigenes. These unigenes were used for gene functional annotation, and seven widely used databases included nr (NCBI non-redundant protein sequences), nt (NCBI nucleotide sequences), Pfam (protein family), KOG/COG (Cluster of Orthologous Groups of proteins), Swiss-Prot (manually annotated protein sequences), KEGG (Kyoto Encyclopedia of Genes and Genome), and GO (Gene Ontology). 

### 2.2. Insect Rearing

To test the knockdown efficiencies of dsRNAs, we used field-collected individuals from outbreak populations in Idaho (Owyhee County: 43°10′59.7″ N 116°39′23.3″ W and 43°16′36.8″ N 116°40′23.1″ W) on 8–9 May 2019, which were collected by the USDA Rangeland Unit and shipped in insulated biomailer boxes to the Department of Entomology at Texas A&M University under a USDA APHIS PPQ permit (P526P-19-01373). Additional specimens were collected by the USDA Rangeland Unit from the same outbreak populations and sent on 2 June 2019. Upon arrival, the insects were sorted by sex into gasket-sealed plastic rearing containers, measuring 61 cm × 45.7 cm × 38.7 cm. The lid and back side were modified with a mesh screen for ventilation, and the bottom was modified with a larger mesh screen to allow frass to fall through. Each rearing container included egg cartons for extra vertical space, and the insects were fed Romaine lettuce daily, a granivorous mixture of Cheerios brand cereal, sunflower seeds, fish flakes, Fluker’s Cricket diet, and wheat bran (1:1:1:1:2 ratio). The insects were kept in a walk-in environmental chamber at a 14:10 day/night cycle at 24 °C and under 40% humidity. Final instar nymphal insects were used for the experiments.

The time course and mortality assays were performed at the USDA APHIS PPQ Field Operations building in Boise, Idaho. Adult Mormon crickets from large migratory bands were collected by sweep nets and by hand from Cow Creek Road, Jordan Valley, Idaho (43°06′24.1″ N 117°00′01.3″ W) on 11 July 2019. These insects were transported to the USDA facility and sorted by sex into large mesh cages (60.2 cm × 36 cm × 36 cm), where the insects were kept at room temperature. For each assay, the insects were isolated into 16 oz. deli cups post-injection of dsRNA and fed daily on a diet of Romaine lettuce and wheat bran. The experiments ran between 11–19 July 2019.

### 2.3. Target Gene Selection

To develop RNAi for Mormon crickets, orthologs for housekeeping genes, constitutively expressed genes with important cellular functions, from the Central American locust *Schistocerca piceifrons* [57], were searched in the Mormon cricket transcriptome using a BLAST search within Geneious (R10.2.6; Biomatters Ltd., Auckland, New Zealand) with a max Expect (E)-value of 10^−5^. The sequences that had the highest pairwise similarities and query coverage were selected, and the search results were compared against the NCBI nr nucleotide database using another BLAST search to verify the identities and functions of the Mormon cricket genes. A total of 11 target genes were initially selected for this study (Table 1): Actin 5C (*Act5C*), Tubulin A1 (*Tub*), Elongation factors 1 alpha and 2 (*EF1a* and *EF2*), Ribosomal protein L32 and L5 (*RpL32* and *RpL5*), Annexin IX (*Ann*), Armadillo (*Arm*), Succinate dehydrogenase (*SDH*), Glyceraldehyde-3P-dehydrogenase (*GAPDH*), and Heat shock protein 70 (*Hsp70*). Of these, we developed dsRNA for seven genes (*Tub*, *EF1a*, *RpL32*, *RpL5*, *SDH*, *GAPDH*, and *Hsp70*).

### 2.4. dsRNA Construction

The sequences of the seven target genes were imported into Geneious, and the coding sequences (CDS) for each sequence were identified. The primers for dsRNA production were designed based on these CDS with Primer3 version 0.4.0 (http://bioinfo.ut.ee/primer3-0.4.0/) [58,59] using the following parameters: 400–600 bp product size, primer size 18–25 bp (optimum: 20 bp), GC clamp set to 1, and optimal melting temperature of 60. Primers for Green Fluorescent Protein (*GFP*), which was used as a control, were previously published [60]. A T7 virus promotor sequence “taatacgactcactatagggaga” was added to the 5′ ends of both forward and reverse primers to ensure compatibility with the MEGAscript™ RNAi Kit (Invitrogen, Waltham, MA, USA) (Table 2). All primers were purchased from Integrated DNA Technologies, Inc., Coralville, IA, USA.

As a template for dsRNA synthesis for the seven target genes, we used pooled RNA from the three females used for RNA sequencing, which was diluted to 100 ng/µL and subsequently converted to cDNA using an iScript cDNA synthesis kit (Bio-Rad, Hercules, CA, USA), following the manufacturer’s guidelines. For the generation of ds*GFP*, GFP-expressing plasmid DNA (Altogen biosystems, Las Vegas, NV, USA) was used as a template. Subsequently, a first PCR was set up using Platinum Taq Polymerase (Invitrogen) on a Bio-Rad T-100 thermal cycler with the following cycle profile: 94 °C for 2 min; 10 cycles of 94 °C for 30 s, 60 °C for 30 s, 72 °C for 1:30 min; 25 cycles of 94 °C for 30 s, 69 °C for 30 s, 72 °C for 1:30; and a final extension at 72 °C for 2 min. The resulting amplicon was used as a template for a second PCR to further increase the dsRNA template concentration. The second PCR was set up as the first PCR, with the following cycle profile: 94 °C for 2 min; 35 cycles of 94 °C for 30 s, 69 °C for 30 s, 72 °C for 1:30 min; and 2 min at 72 °C. The resulting amplicon was concentrated using a Monarch PCR Cleanup Kit (NEB, Ipswich, MA, USA) following standard procedures. The products varied in size but were within the 300–600 bp range (Table 2); this is consistent with previous work on efficient dsRNA primer design [23]. The final product was used as a template for generating dsRNA using the MEGAscript™ RNAi Kit (Invitrogen), following the manufacturer’s guidelines. Each dsRNA was quantified with a Denovix DS-11 spectrophotometer and diluted to 20 ng/µL in locust saline solution (1 L: 8.766 g NaCl; 0.188 g CaCl_2_; 0.746 g KCl; 0.407 g MgCl_2_; 0.336 g NaHCO_3_; 30.807 g sucrose; 1.892 g trehalose; pH 7.2).

### 2.5. qPCR Primer Design

qPCR primers were designed for all 11 genes included in the study, using Primer3 version 0.4.0. (http://bioinfo.ut.ee/primer3-0.4.0/) as described above, but with amplicon size set to 75–200 bp. To avoid false positives due to amplifications of siRNA’s, qPCR primers for target genes were generated outside of the gene region targeted by the dsRNA primers. All primers were purchased from Integrated DNA Technologies, Inc. To assess primer efficiency, the pooled RNA described above was converted to cDNA using the iScript cDNA synthesis kit (Bio-Rad, Hercules, CA, USA), as described above. Primer efficiency was assessed with a 5-fold cDNA dilution series with the following dilutions: 1/1, 1/5, 1/25, 1/125, 1/3125, 1/6125. Each primer pair was tested with qPCR, using the dilution series as a template. Gene expression was assayed by applying 2.5 μL of cDNA template to 5 μL SYBR green supermix (Bio-Rad, Hercules, CA, USA) and 1.25 μL each of forward and reverse primers. The reaction was run in a Bio-Rad CFX-384 qPCR thermal cycler under the following setting: 3 min at 95 °C, followed by 39 cycles of 10 s at 95 °C and 30 s at 60 °C. A melt curve was also included. Each sample was tested with primers for the target gene and two housekeeping genes. The primer efficiency was obtained by plotting the C_q_ value to the logarithm of the dilution.

### 2.6. RNAi Knockdown Efficiency

We used a dose of 120 ng of dsRNA per injection (6 μL of 20 ng/μL dsRNA) for all experiments, which is well above the amount described by Wynant et al. [41], who found that application of as little as 15 pg of dsRNA per mg of tissue is sufficient to completely silence specific gene expression in adult desert locusts. The test subjects were injected through the membrane between the second and third abdominal sternite on the right side with a Hamilton precision microinjection pump (700 series, 705RN, 50 µL, Sigma-Aldrich, St. Louis, MO, USA) with a 22-gauge needle sterilized with acetone, ethanol, and locust saline solution between injections. We tested knockdown efficiencies for the dsRNA of all seven genes. Ten sixth-instar females were injected with dsRNA for each target gene or *GFP* as a control. After the injection, the insects were isolated in individual 16 oz. deli cups and fed Romaine lettuce and wheat bran for 24 h, after which the head and thorax tissue were dissected, snap frozen in liquid nitrogen, and preserved at −80 °C. RNA extraction and cDNA production were performed as described above, and the mRNA level was quantified using qPCR. We used *EF1a* and *Ann* (and *Act5C* when testing *EF1a*) as reference genes for qPCR. To assess the knockdown efficiencies, 10 replicates of head tissue, and 5 replicates of thorax tissue per target gene were used.

### 2.7. Time-Course and Mortality Assays

The time course assay was designed to quantify gene knockdown at the mRNA level after injection for seven days. The mortality assay was designed to characterize how long it would take for insects to die after injecting the dsRNA of target genes. For both assays, 35 individuals were used for each of the four target genes (*EF1a, GAPDH, Hsp70, RpL5*) and the *GFP* control. Thus, a total of 350 individuals were used. Each test subject was injected with 10 μL of dsRNA at a concentration of 20 ng/μL (the volume was increased because the test subjects were adults, which were larger than nymphs) and given a booster injection of the same quantity and concentration three days after the initial injection. To quantify the time course of gene knockdown, five individuals from each of the five gene treatments were killed 24 h after the initial injection, and the head tissues were dissected and preserved in an RNALater (Ambion) solution. This process continued at the same time every day for seven days. RNA extraction, cDNA production, and qPCR were performed as described above. To assess RNAi-induced mortality, the number of insects that were alive after injection was counted each day for seven days. Any other signs (lethargic behavior, curled legs, or individuals that lay on their backs or sides) were also noted.

### 2.8. dsRNA Degradation Assay

To test for the presence of dsRNA-degrading enzymes in the hemolymph, we followed the protocol described by Singh et al. [61]. In short, we collected hemolymph from live insects by puncturing the hind coxal membrane with a small insect pin. 300 μL of hemolymph per sample was collected into 1.5 mL plastic vials containing 0.05 mg/mL phenylthiourea dissolved in methanol (0.05 mg/mL) and kept on ice to prevent melanization. Hemocytes were removed by centrifugation at 13,000 rpm for 10 min at 4 °C, and the resulting supernatant was serially diluted. 100 ng of ds*GFP* was incubated in the serially diluted hemolymph samples for 1 h, and the resulting samples were subjected to electrophoresis on a 1% agarose gel. The resulting gel was viewed under UV light and imaged.

### 2.9. Statistical Analysis

All qPCR data were analyzed using the ΔΔC_q_-method and *p*-values were derived from Student’s *t*-test or one-way analysis of variance (ANOVA) at significance levels of 0.05 (in R, Version 3.6.1). All groups were subjected to normality and variance assumptions. The groups that did not meet the normality assumptions were subjected to the Wilcoxon test (R, Version 3.6.1). All groups had at least three biological and two technical replicates for analysis.

## 3. Results

### 3.1. Transcriptome

Illumina sequencing produced raw reads that were of high quality, with very low error rates (0.01–0.02%). On average, ~88 million raw reads (82.9 million clean reads or 12.42G of clean bases) were produced for each library. After de novo assembly, the total number of unigenes was 317,335 with an N50 of 1282 bp (Supporting Information). These unigenes were functionally annotated using the seven databases, but the percentage of successful annotation was low overall, with 44.07% of the unigenes annotated in at least one database (Supporting Information). The raw reads have been submitted to the NCBI (SRA numbers: SRS9867025-SRS9867033) under BioProject PRJNA756613. This assembled transcriptome has been deposited at DDBJ/EMBL/GenBank under the accession GJIY00000000. The version described in this paper is the first version, GJIY01000000.

### 3.2. dsRNA Knockdown Efficiency

We identified orthologs of seven housekeeping genes in the Mormon cricket, and used these as targets for RNAi. These genes are abundantly expressed across tissues and have important cellular functions [62], making them optimal targets for testing RNAi in this organism. We observed gene-specific and tissue-specific RNAi responses 24 h after injection. In the head tissue, dsRNA injection induced a strong knockdown in ds*EF1a* (8-fold change), ds*GAPDH* (13-fold change), and ds*SDH* (7-fold change), and a moderate knockdown in ds*Hsp70* (2-fold change) (Figure 2). There was no knockdown observed for ds*RpL5,* ds*RpL32*, and ds*Tub* (<2-fold change). In the thorax tissue, we observed a trend toward overexpression in ds*RpL32* (2-fold change), and ds*SDH* (2-fold change), a strong knockdown in ds*GAPDH* (32-fold change), and a moderate knockdown in ds*RpL5* (2-fold change) (Figure 2). There was no knockdown observed in ds*EF1*, ds*Hsp70*, and ds*Tub* (<2-fold change).

### 3.3. Time-Course Assay of RNAi Response

The time courses of head mRNA expression levels after dsRNA injection and a booster injection showed clear differences for the different target genes (Figure 3). For ds*EF1a*, there was no knockdown observed for the first 48 h after injection, but from the 4th day, we observed a significant knockdown (3- to 16-fold change), which continued over the course of seven days (Figure 3). Injection of ds*GAPDH* resulted in a significant knockdown (7-fold change) after 24 h, but *GAPDH* expression recovered to twice that level on days 2 and 3 (Figure 3). After the booster injection, *GAPDH* expression levels alternated between reduction and recovery. *Hsp70* was overexpressed after 24 h, and did not show any significant knockdown until the 4th day, followed by a recovery on day 6 (Figure 3). *RpL5* had a significant knockdown (3- to 9-fold change) from day 1, which continued throughout the whole 7-day experiment (Figure 3).

### 3.4. Mortality Assay

Over the course of seven days, only three of 175 tested insects died. These included one individual injected with ds*EF1a* that died 24 h after the initial injection, one individual injected with ds*GFP* that died on day 3, and another from the same group that died on day 5. The remaining insects did not show any signs of weakness or lethargy.

### 3.5. dsRNA Degradation Assay

We observed that ds*GFP* was severely degraded when incubated in the hemolymph for 1 h (Figure 4). The effect of degradation decreased with the increase in dilution, but even at the 1:160 dilution, the effect of dsRNA degradation was noticeable (Figure 4).

## 4. Discussion

This study represents the first attempt at developing RNAi for the Mormon cricket by generating a de novo transcriptome, which allowed for the identification of housekeeping genes that were used as target genes for gene knockdown by RNAi. We have demonstrated that it is possible to elicit RNAi responses in Mormon crickets by injection, but knockdown efficiencies varied according to target genes and tissue types (Figure 2). The time course of the RNAi response also varied according to the target genes (Figure 3). We also show that one of the reasons for the poor knockdown efficiencies could be the presence of dsRNA-degrading enzymes in the hemolymph (Figure 4).

For an insect pest species for which genomic resources are lacking, de novo transcriptomics is an attractive option for enabling RNAi development. Although it is ideal to start with a reference genome when developing any tools for functional genetics, the availability of such genomic resources varies across taxa and is particularly scarce for Orthoptera. This is because orthopterans are known to have large genome sizes [63] and, in fact, the largest genomes among insects, which are challenging and costly to sequence. Currently, there is no genome available for the Mormon cricket, which is estimated to have a large genome with a roughly 6.8 Gb in size (Song et al. unpublished), although a collaborative effort with the USDA to sequence its genome is nearing completion. Thus, as an alternative way of generating genomic resources for RNAi development, we relied on de novo transcriptome sequencing for the Mormon cricket. Based on the transcripts and a BLAST search, we were able to identify orthologs for important housekeeping genes for qPCR primer and dsRNA development. Because housekeeping genes are abundantly expressed across tissues [62], we found de novo transcriptomics to be a good compromise as a means to develop genomic resources for a non-model organism.

Although oral delivery has not been shown to be effective for RNAi in orthopterans, including locusts [64], it is known to work very well via a simple injection of dsRNA into the hemolymph, which causes systemic knockdown [41]. Thus, our initial expectation was that RNAi would work equally well for Mormon crickets. To our surprise, however, the knockdown efficiencies varied according to the target genes and tissue types in the Mormon crickets. For example, in the head tissue, the knockdown efficiency varied wildly among the housekeeping genes, such that dsRNAs of some genes (*EF1a, GAPDH*, and *SDH*) yielded knockdowns of 80% or greater, while those of other genes resulted in no knockdown or even overexpression. The same pattern was true for the thorax tissue, except that the resulting knockdown pattern was different from that of the head tissue. Only ds*GAPDH* consistently causes significant knockdown in both tissues out of the seven housekeeping genes tested (Figure 3). Of those dsRNAs that yielded knockdown, not a single one resulted in a highly efficient knockdown (e.g., 99%), levels of which have commonly been found in locusts [41,42,65,66,67].

Variable efficiency of the RNAi response can be caused by multiple non-mutually exclusive factors [68], including inappropriate selection of target genes and poor design, unusual tissue-specific responses, variable physiological conditions, as well as the presence of extracellular nucleases. Although any one of these could explain the patterns we observed in the Mormon crickets, we suspect that some factors are more important than others. Our finding that the RNAi responses differed between the head and thorax tissues is consistent with previous studies showing tissue-dependent RNAi efficiencies in various insects, including locusts [41], mosquitoes [69], and moths [70]. The causes of the tissue dependency of the RNAi response are unclear, but it is possible that dsRNA uptake mechanisms could vary across different organs and tissue types [41]. For example, Miller et al. [71] reported in *Drosophila* larvae that many types of tissues lack an uptake mechanism for dsRNA. We administered our dsRNAs to the abdomen and quantified the knockdown efficiencies in the head and thorax tissues. Differences in the expression patterns of the same genes between the two tissue types potentially highlight discrepancies in maintenance needs between different tissue types. Even though housekeeping genes are expressed in every cell and are constantly transcribed, the level of transcription may not necessarily be the same for different cell groups [62].

The overexpression of a gene by dsRNA is sparsely reported in the literature [72,73], and most of them are hormone- or immune-related genes [74,75,76], but we observed this pattern in *SDH* and *RpL32* in the thorax tissue (Figure 2). *SDH* is an enzyme found in the inner mitochondrial membrane and is involved in the electron transport chain [77], and *RpL32* is a component of the ribosomes. It is unclear why RNAi targeting these genes resulted in significant overexpression, but we can speculate that the RNAi machinery might create a stressful environment for the cell by suppressing its own gene expression instead of a foreign agent’s genes. The knockdown efficiency assay data were collected 24 h after injection, so the overexpressed gene may be a snapshot of the cell attempting to upregulate transcript production to compensate for the dearth of proteins required for maintenance.

Therefore, it was important to characterize the time course of the RNAi response beyond the first 24 h after injection. Of the four genes we characterized on the head tissue, only *dsRpL5* showed gradual and persistent knockdown over the 7-day period (Figure 3). The reason that other genes did not show the persistent pattern could be that these housekeeping genes were already abundantly transcribed at the time of injection, so there was a lag in terms of inducing strong RNAi responses over time. ds*Hsp70* initially caused overexpression and resulted in oscillation between moderate knockdown and overexpression over the duration of the experiment (Figure 3). This pattern could be due to the conflict between dsRNA-induced transcript degradation and transcriptional upregulation in a stressful cellular environment caused by dsRNA. Heat shock proteins (HSPs), including *Hsp70*, exhibit chaperone-like behavior, helping denatured proteins refold during times of stress [78,79]. As the RISC degrades transcripts of *Hsp70* (and possibly other genes from homologous siRNA’s generated by DICER), the cell could experience a stressful environment as it is attacked by its own machinery, which could form a feedback loop to upregulate transcription of more HSPs.

The initial injection of dsRNA for *GAPDH* seemed to have worked well in reducing expression, but the booster injection on day 4 seemed to induce an alternating pattern of severe knockdown and recovery (Figure 4). Given that *GAPDH* is involved in many different functions besides metabolism, such as transcription, DNA repair, apoptosis, and membrane trafficking [80], it is highly involved in many cellular regulatory processes. *GAPDH* knockdown might have a similar cellular scenario to *Hsp70* knockdown, where the cell is caught in RNAi-induced stress due to a deficiency in *GAPDH*, so the insect could have upregulated its expression to return to homeostasis. It would be important for future studies to also look at protein expression in tandem with transcript expression levels since the two may not necessarily be proportional.

There was a difference between the first knockdown efficiency results and the initial expression level from the time course results for each candidate gene. This could be due to differences in dosage and life stage used. We used adults and a higher dosage for the time-course assays, while penultimate instars were used for the knockdown efficiency assays. Differences in dsRNA responses have been recorded in different life stages of the same insect, possibly due to differences in their physiological needs [81].

One of the most common reasons for the low knockdown efficiency known in insects is the presence of extracellular nucleases or dsRNases, which essentially neutralize dsRNAs [82]. RNA efficiency can vary among different insect orders and even between closely related groups [21,34,35,61,68]. Many beetles are considered RNAi sensitive due to the low doses needed to elicit the RNAi response [21], but many Lepidopterans are recalcitrant for RNAi, thereby requiring large doses to achieve some knockdown [70]. It is largely suspected that RNAi is ineffective in Lepidoptera due to the presence of dsRNases, which break down dsRNAs prior to cellular uptake [70,83]. dsRNases could still be present even though some of the dsRNAs were successful; the endonuclease DICER will cleave any dsRNA into siRNA’s for the RISC to use so partially degraded dsRNAs can still lead to some degradation. For Orthoptera, Singh et al. [61] showed that a cricket (*Gryllus texensis*) had higher dsRNase activity in the body fluid than a grasshopper (*Syrbula admirabilis*). Initially, we did not suspect strong dsRNase activity in the hemolymph of the Mormon crickets because dsRNA injection works so well in locusts, which implies that the amount of dsRNases in the hemolymph of the locusts is negligible [84]. In the desert locust, the dsRNase level is high in the midgut but not so in the hemolymph, making it easy to cause systematic knockdown via injection [41,85]. Despite our efforts to silence all seven housekeeping genes in the Mormon cricket, we did not achieve highly efficient knockdown (95%) for any single gene. Furthermore, our dsRNAs did not induce any mortality. Collectively, these findings led us to suspect that the hemolymph of the Mormon crickets could harbor extracellular nucleases that might digest dsRNAs. In fact, our degradation assay using hemolymph alone clearly showed high dsRNase activity in the Mormon cricket (Figure 4). Therefore, we infer that the poor RNAi efficiency could be partially due to the presence of dsRNase in the hemolymph.

## 5. Conclusions

In this study, we developed tools to successfully generate RNAi silencing in Mormon cricket. Though our dsRNAs varied in their efficiencies and none of the genes we tested could induce high mortality in these insects, our results demonstrate that there is strong dsRNase activity in the hemolymph of the Mormon crickets, which was previously unknown, and may be the reason for the lack of consistent knockdown and mortality. There is clearly more work that needs to be done to improve delivery and efficacy before RNAi can be used as a population management tool for Mormon crickets, but the present work represents a key first step toward this goal, and more informed efforts are underway.

## Figures and Tables

**Figure 1 insects-13-00739-f001:**
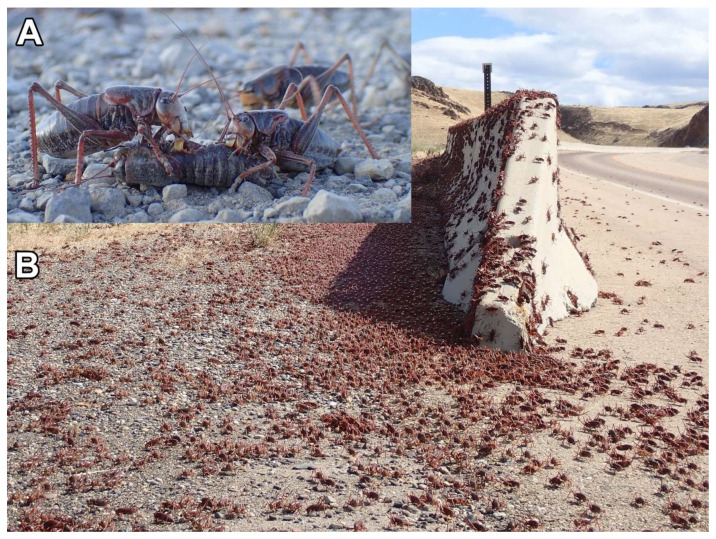
(**A**) Mormon crickets cannibalizing each other during a population outbreak in Idaho in 2020; (**B**) A population outbreak of Mormon crickets along a major highway in Idaho in 2019. (Photo credits: (**A**): H.S.; (**B**): D.A.W.).

**Figure 2 insects-13-00739-f002:**
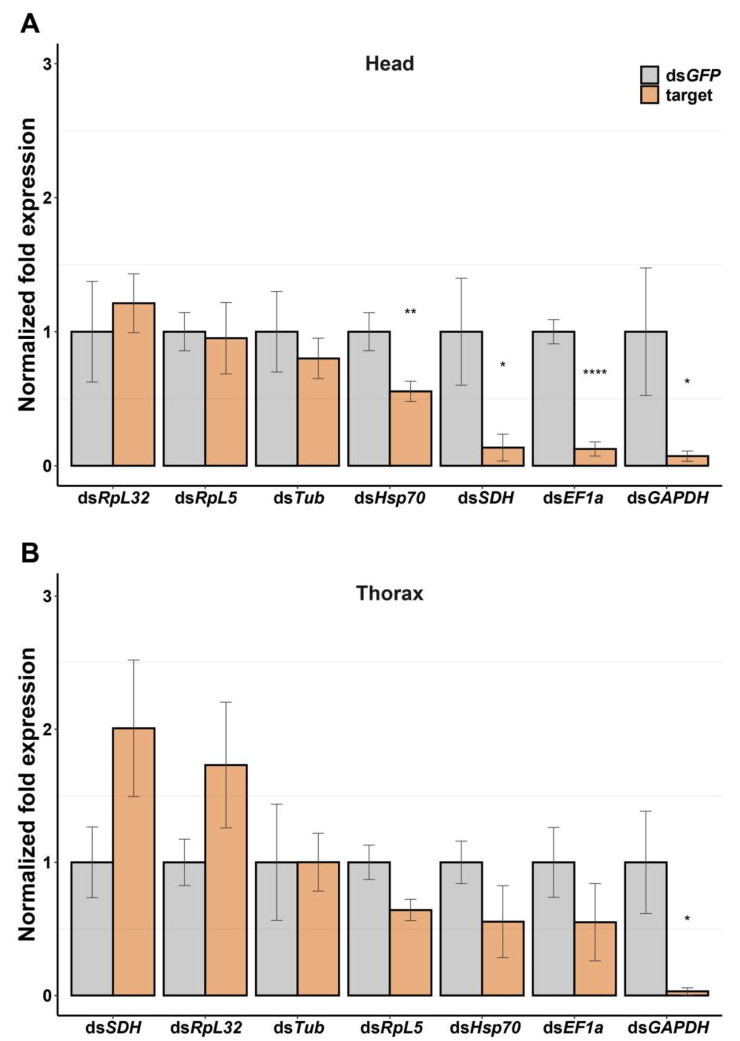
Knockdown efficiency assay in different tissues. (**A**) Head tissue, (**B**) Thorax tissue. Each gene tested had a sample size of at least 5. Error bars indicate the standard error of the mean for each control (*GFP*) and target gene. Data were analyzed using the ΔΔCt method. The statistical significance of differences was analyzed with the Student’s t-test or Wilcoxon test. Significant differences between the control (*GFP*) and target gene are indicated by asterisks (*: *p* < 0.05; **: *p* < 0.005; ****: *p* < 0.0001).

**Figure 3 insects-13-00739-f003:**
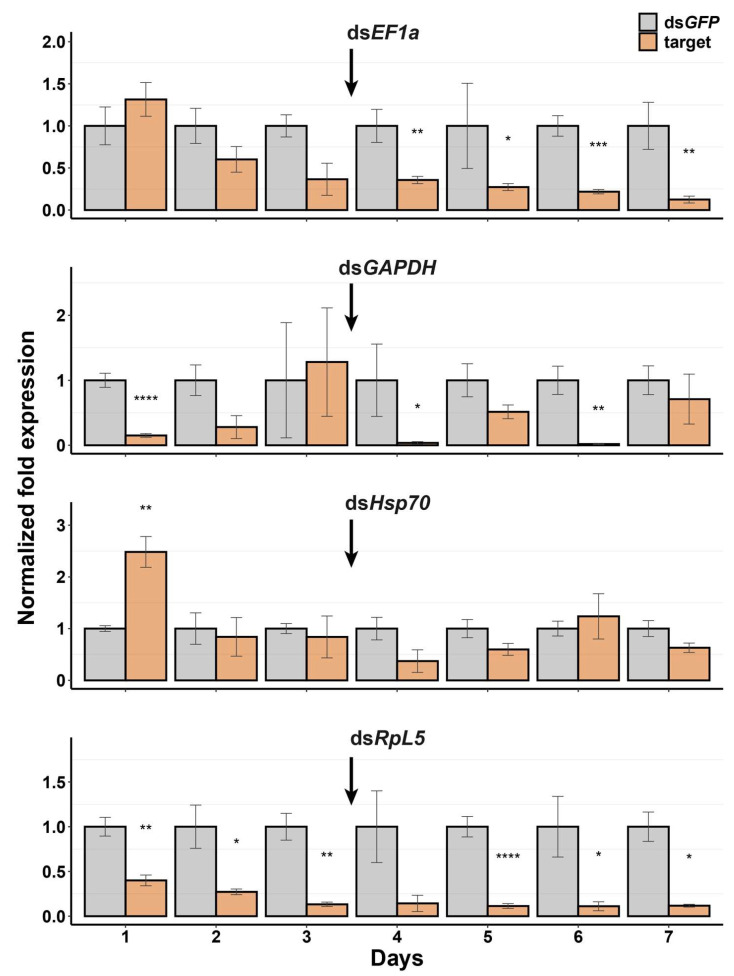
Time course assay for *EF1*, *GAPDH*, *Hsp70*, and *RpL5* target genes. Each gene per day had at least a sample size of 3. Error bars indicate the standard error of the mean for each control (*GFP*) and target gene. Data were analyzed using the ΔΔCt method. The arrow indicates the booster injection. The statistical significance of differences was analyzed with the Student’s *t*-test or Wilcoxon test. Significant differences between the control (*GFP*) and target gene are indicated by asterisks (*: *p* < 0.05; **: *p* < 0.005; ***: *p* < 0.0001; ****: *p* < 0.00001).

**Figure 4 insects-13-00739-f004:**
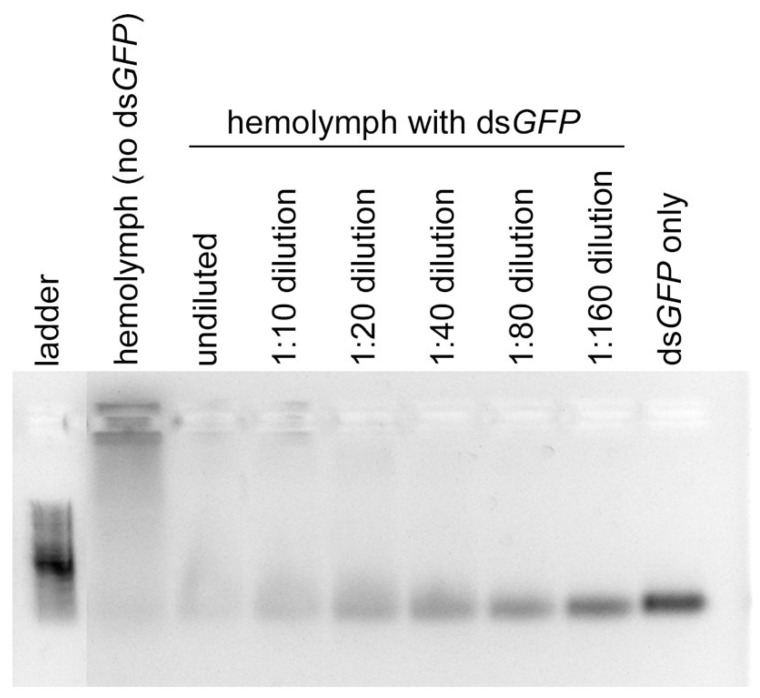
dsRNA degradation assay for hemolymph. dsRNA completely degrades in undiluted hemolymph, strongly suggestive of high dsRNase activity.

**Table 1 insects-13-00739-t001:** Primers used for qPCR. Melting temperature (Tm) and primer efficiency (E) are listed for each qPCR primer set.

Gene Name	Abbreviation	Primer Sequence	Tm (°C)	E (%)	Genbank Accession
Actin 5C	*Act5C*	F: CACCCTCAAGTACCCCATTG	55.4	94.29	ON402773
		R: GTCAGCAGGATTGGGTGTTC			
Annexin IX	*Ann*	F: CAATTTTGGTGACCCGTAGC	54.3	93.00	ON402772
		R: CAATAGCCAGCAGTCCCTTC			
Armadillo	*Arm*	F: TCCTGGCAATTGTGACAGAC	55.2	101.48	ON402771
		R: ATTCGTACCAGCTCCACAGG			
Elongation Factor 1Alpha	*EF1a*	F: CAAGATGGGCTGGTTTAAGG	53.6	94.50	ON402770
		R: CTCAGTAGGCCTGGAAGGTG			
Elongation Factor 2	*EF2*	F: GCAAACCGAAACTGTCCTTC	54.4	100.75	ON402769
		R: TGACGTTCTCCACAATACGC			
Glyceraldehyde-3P-Dehydrogenase	*GAPDH*	F: TACTCATGGCCGCTTCAAGG	57.4	99.37	ON402768
		R: GGAATGCTTTTGGGGTCACG			
Heat Shock Protein 70	*Hsp70*	F: TTTTGGACAAGTGCAACGAG	53.8	92.63	ON402767
		R: AATGATGGGGTTGCAAAGAG			
Ribosomal Protein L5	*RpL5*	F: GGTGCCAGAGTGTTTGGTG	53.2	96.40	ON402766
		R: ACTCTTTTGATTCCGCATCG			
Ribosomal Protein L32	*RpL32*	F: GTTGGTGCACAATGTGAAGG	54.8	97.80	ON402765
		R: CCACGATAGACTTCCGCTTC			
Succinate Dehydrogenase	*SDH*	F: CCCTAGAGAAGTAGAGGCTGC	56.3	101.00	ON402764
		R: CCCAGCTCCATTGACCAGAC			
Tubulin A1	*Tub*	F: AACAGCTTATCACGGGCAAG	55.5	97.50	ON402763
		R: GCTTTCTGATGCGATCCAAG			

**Table 2 insects-13-00739-t002:** dsRNA primers used in this study and product size.

dsRNA	Primer Sequence	Product Size
ds*EF1a*	F: taatacgactcactatagggagaAATATGCCTGGGTGTTGGAC	479
	R: taatacgactcactatagggagaATCCCTTAAACCAGCCCATC	
ds*GAPDH*	F: taatacgactcactatagggagaAATCAAGTGGGGAGCTGATG	314
	R: taatacgactcactatagggagaCAGTGCTTGCAGGAATGATG	
ds*Hsp70*	F: taatacgactcactatagggagaTGATGCAGCAAAGAACCAAG	336
	R: taatacgactcactatagggagaGGCTCCAGCATCCTTTGTAG	
ds*RpL5*	F: taatacgactcactatagggagaCCTCCGTCTGATCTCTCAGG	321
	R: taatacgactcactatagggagaTCCCCGACCACTTTCTACAG	
ds*RpL32*	F: taatacgactcactatagggagaGAAGCGCAATAAGCACTTCG	303
	R: taatacgactcactatagggagaCCACGATAGACTTCCGCTTC	
ds*SDH*	F: taatacgactcactatagggagaAATTTTCGATCTGGGTGGTG	388
	R: taatacgactcactatagggagaTTTTTGCACCTTCGGGATAC	
ds*Tub*	F: taatacgactcactatagggagaAACAGCTTATCACGGGCAAG	469
	R: taatacgactcactatagggagaCCATCAAACCGAAGAGAAGC	
ds*GFP*	F: taatacgactcactatagggagaACGTAAACGGCCACAAGTTCAGC	N/A
	R: taatacgactcactatagggagaGAGGGTCTTCTGCTGGTAGTGGTCG	

## Data Availability

The raw reads have been submitted to the NCBI (SRA numbers: SRS9867025-SRS9867033) under BioProject PRJNA756613. This assembled transcriptome has been deposited at DDBJ/EMBL/GenBank under the accession GJIY00000000. The version described in this paper is the first version, GJIY01000000.

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
