# Peer review of "Development of RNAi Methods for the Mormon Cricket, Anabrus simplex (Orthoptera: Tettigoniidae)"

_insects, 2022, doi:10.3390/insects13080739_

Round 1
Reviewer 1 Report
Dear Authors,
I enjoyed reading your manuscript, Development of RNAi methods for the Mormon cricket, Anabrus simplex (Orthoptera: Tettigoniidae), submitted to Insects. This important study is well written and the conclusions are justified by the Results. I am happy to suggest that the paper be accepted in its current form, and only found one small typo that can be addressed at proofing:
Simple Summary, and Abstract: Change the beginning, to correct the singular/plural issue: "The Mormon crickets (Anabrus simplex) are native" should change to "The Mormon cricket (Anabrus simplex) is native..", or otherwise lose "the" (like at the beginning of the abstract) to just "Mormon crickets are a major rangeland pest..."
Congratulations again on a great study!
Darron Cullen
*ends*
Author Response
We would like to thank the reviewer for the positive review. The reviewer requested a very minor change in the abstract, which we have fixed.
Reviewer 2 Report
The manuscript investigated whether RNAi can be used as an alternative control method for Mormon cricket. The authors conducted RNA-Seq analysis on head, thorax, and abdomen, injected dsRNAs targeting seven housekeeping genes, and determined gene expression and mortality. They found varying efficiencies of gene silencing for target genes and dsRNA degradation activity in the hemolymph. Overall, the experiments were conducted well, and this manuscript's findings are enough to support the main conclusion. I have a few suggestions to improve the manuscript.
Since de novo transcriptome data is available, why didn’t the authors report tissue-specific expression data for seven target genes? Although targeted genes were housekeeping genes, their expression patterns of them might differ as the authors described in L431-435. These data would be important to understand tissue-specific knockdown in response to dsRNA treatment. Please include these expression data.
Why didn’t the authors measure gene expression in abdomen? It would be expected to show higher gene silencing efficiency in abdomen where dsRNA was injected than in other tissues such as head and thorax.
L19:RNAi has been studied for pest management, but none or very few products are available on the market. Seeds of SmartStax® PRO with RNAi Technology are available in 2022 and several companies are conducting dsRNA-based pesticide trials. It would be appropriate to say that RNAi has potential for pest management. L19 could imply that RANi is already used in the fields.
Constructs could mean plasmids that express dsRNA. This study used in-vitro synthesized dsRNA. It would be better to describe “dsRNA”, rather than “dsRNA constructs”.
L248: change Table 1 to Table 2
Please provide qPCR conditions such as the amount of cDNA, PCR condition, melting curve analysis, etc.
L339: The expression of Rpl5 was not significantly different from dsRl5 and dsGFP, indicating there was no moderate knockdown.
L340: change dsTUB to dsTub
dsEF1a, dsGAPDH, and dsRpl5 showed persistent knockdown for seven days, up to 80-90% knockdown, even though there is dsRNA degradation activity in the hemolymph. However, no mortality was observed. It is often observed that 90% gene silencing of housekeeping genes leads to reduced growth or death in insects. Would it be because the presence of other homologs could compensate gene silencing effect of those genes? Or are those genes not suitable for controlling Mormon crickets?
Fig. 4, the size of the ladder was not visible. Please provide a better gel picture.
Author Response
We would like to thank the reviewer 2 for providing thoughtful comments. We were able to address the comments below. Our responses are in blue.
The manuscript investigated whether RNAi can be used as an alternative control method for Mormon cricket. The authors conducted RNA-Seq analysis on head, thorax, and abdomen, injected dsRNAs targeting seven housekeeping genes, and determined gene expression and mortality. They found varying efficiencies of gene silencing for target genes and dsRNA degradation activity in the hemolymph. Overall, the experiments were conducted well, and this manuscript's findings are enough to support the main conclusion. I have a few suggestions to improve the manuscript.
Thank you so much for the positive comments.
Since de novo transcriptome data is available, why didn’t the authors report tissue-specific expression data for seven target genes? Although targeted genes were housekeeping genes, their expression patterns of them might differ as the authors described in L431-435. These data would be important to understand tissue-specific knockdown in response to dsRNA treatment. Please include these expression data.
The transcriptome data were generated from field-collected specimens for the sake of generating as much transcripts as possible, and thus were not particularly suitable for generating any expression data. Based on the housekeeping genes we identified from the transcriptomes, we did quantify the expression level when calculating the primer efficiency, all genes were abundantly expressed in both issues, as expected from any housekeeping genes. Thus, we did not pursue further to characterize the expression patterns of these genes further.
Why didn’t the authors measure gene expression in abdomen? It would be expected to show higher gene silencing efficiency in abdomen where dsRNA was injected than in other tissues such as head and thorax.
Our assumption was that RNAi would be systemic in Mormon crickets. Thus, the effect of RNAi would spread (and we show that it did) to head and thorax. We did not measure gene expression in abdomen for practical reasons. Mormon crickets tend to have a very large abdomen, which makes it costly to RNA extraction, and since we were able to show the patterns in two tissues (head and thorax), we did not think it would be critical to include the abdomen as a third tissue.
L19:RNAi has been studied for pest management, but none or very few products are available on the market. Seeds of SmartStax® PRO with RNAi Technology are available in 2022 and several companies are conducting dsRNA-based pesticide trials. It would be appropriate to say that RNAi has potential for pest management. L19 could imply that RANi is already used in the fields.
Thanks for this comment, and we changed the text accordingly.
Constructs could mean plasmids that express dsRNA. This study used in-vitro synthesized dsRNA. It would be better to describe “dsRNA”, rather than “dsRNA constructs”.
Thanks for this comment and we changed throughout the text to avoid any confusion.
L248: change Table 1 to Table 2
Thanks for catching this mistake and we changed it accordingly.
Please provide qPCR conditions such as the amount of cDNA, PCR condition, melting curve analysis, etc.
Thanks for this comment. We included the qPCR parameters and conditions.
L339: The expression of Rpl5 was not significantly different from dsRl5 and dsGFP, indicating there was no moderate knockdown.
We think that the reviewer was mistaken. The expression of Rpl5 was moderately knockdown in the thorax tissue, but not in the head tissue. We believe that the reviewer was looking at the graph for the head tissue.
L340: change dsTUB to dsTub
Thanks for catching this mistake and we changed it accordingly.
dsEF1a, dsGAPDH, and dsRpl5 showed persistent knockdown for seven days, up to 80-90% knockdown, even though there is dsRNA degradation activity in the hemolymph. However, no mortality was observed. It is often observed that 90% gene silencing of housekeeping genes leads to reduced growth or death in insects. Would it be because the presence of other homologs could compensate gene silencing effect of those genes? Or are those genes not suitable for controlling Mormon crickets?
Thanks for this comment. This is an important comment. It was puzzling for us to observe that 90% would not kill the insects. In other insects that we have worked with, this level of knockdown in housekeeping genes would certainly kill the insects. Our study was intended for developing RNAi methods for Mormon crickets, not necessarily to develop lethal RNAi. For that, we would have to characterize a specific target that has a very distinct phenotypic effect. For this study, we just wanted to test whether RNAi would be possible for this species. It is difficult to know why RNAi does not kill the insects. It is possible that there could be some sort of compensatory actions, or residual proteins that linger on. Rather than being speculative, we have decided to report on our findings.
Fig. 4, the size of the ladder was not visible. Please provide a better gel picture.
Unfortunately, this was the best picture that we had. Since the point of the figure is the degradation patterns of dsGFP, the poor quality of ladder image is not very relevant. So we did not make any further modification to the figure.